# Procedural elements of the complete examination and screening of the healthy term neonate: a protocol for a scoping review and evidence map

Liz M Greene ,[1,2] Rhona O'Connell,[1] Margaret Murphy,[1] Josephine Hegarty[1]

[1]School of Nursing and Midwifery, University College Cork, College Road, Cork, Ireland
[2]School of Nursing, Midwifery and Health Systems, University College Dublin, Dublin, Ireland

**Correspondence to**
Liz M Greene; ElizabethGreene@umail.ucc.ie

## ABSTRACT

**Introduction** All newborns undergo a Complete Examination and Screening of the Neonate (CESoN) to verify the general health and well-being of the neonate and to screen for signs and symptoms of illness and significant congenital disorders, typically within 72 hours of birth. For healthy, term gestation neonates, this examination is usually performed by a qualified healthcare practitioner that is, a midwife, nurse or physician just prior to discharge from the maternity services. As a precursor to modifying and adapting an instrument that measures the quality of performance of the CESoN by healthcare professionals, this review aims to identify, evaluate, synthesise and map the evidence and theory underpinning current practice and the procedural elements of the CESoN.

**Methods and analysis** This review will be guided by the Joanna Briggs Institute methodology for scoping reviews and also the recommendations of the Campbell Collaboration for systematic evidence mapping. Based on the research question, the Person, Concept, Context framework will be used to develop eligibility criteria for inclusion in the review. Eligible information shall be sourced by searching electronic databases including PubMed, Cumulative Index of Nursing and Allied Health Literature, and Scopus, and the published guidance from expert bodies on newborn examination and screening (eg, National Institute for Health and Care Excellence, American Academy of Pediatrics, Royal College of Paediatrics and Child Health) and the grey literature. This study will include primary and secondary research papers, evidence-based guidelines, and expert text and opinions published in English from 2013 to September 2023.

**Ethics and dissemination** Ethical approval is not required for this scoping review and systematic evidence mapping. The results from this study will be disseminated through peer-reviewed format, that is, conference proceedings and peer-reviewed healthcare journals.

## INTRODUCTION
### Rationale

The Complete Examination and Screening of the Neonate (CESoN) is a detailed physical examination of the neonate that is performed within a recommended 72 hours following birth. The CESoN is a separate and more detailed physical examination to

### WHAT IS ALREADY KNOWN ON THIS TOPIC

⇒ The Complete Examination and Screening of the Neonate (CESoN) is an important, universal baseline assessment of newborn health and well-being. Despite its importance, there is a paucity of research on many of the psychomotor and assessment skills utilised by clinicians to conduct the CESoN.

### WHAT THIS STUDY ADDS

⇒ This study hopes to provide clarity on both the sources and levels of evidence underpinning recommendations for optimal performance of the CESoN. This scoping review is part of an evidence mapping process to inform modifications that may be needed to update and pilot an instrument that measures clinician performance of the CESoN.

### HOW THIS STUDY MIGHT AFFECT RESEARCH, PRACTICE OR POLICY

⇒ This study may positively affect practice by providing clinicians and academics with an evidence map that illustrates the strength of evidence underpinning the identified performance elements of the CESoN. As the ultimate goal of the CESoN is to protect neonatal health by detecting early signs and symptoms that may indicate illness or congenital disorders, it is vital that the examination and screening techniques are aligned to the latest and best level evidence available.

that performed within an hour of birth and typically takes no longer than 15 min. The purpose of this detailed physical examination in the healthy term gestation neonate is twofold. The first is to verify the general health and well-being of the neonate prior to discharge from the maternity service. The second is to screen the baby for signs and symptoms of specific and significant congenital disorders, which if not detected within the recommended time frame, may result in severe illness.[1–5] The Newborn and Infant Physical Examination (NIPE) programme[1] describes four of the key screening procedural elements (eyes, heart, hips and testes)

of the CESoN. A summary of all the recommended procedural elements of the CESoN, as per the National Institute for Health and Care Excellence,[2] Public Health England[1] and the American Academy of Pediatrics[3] is provided (see online supplemental file 2 appendix 1). Optimal performance of the procedural elements of the CESoN is important, particularly the screening elements, given the reality that suboptimal technique may result in delayed diagnosis and initiation of treatment for progressive congenital disorders.[1 6 7] A 2023 systematic review[8] identified an instrument[9] developed to measure the quality of performance of the CESoN by qualified healthcare professionals.

## Objectives

As a precursor to modifying and adapting the identified instrument[9] (which was originally developed and piloted the early 2000s), we aim to identify, evaluate and synthesise the evidence and theory underpinning current practice and procedural elements of this detailed clinical examination. The results of this synthesis and evidence mapping will help guide the future development of standardised practice assessment tools for practitioners of the CESoN based on the best available evidence.

The overarching question that we will attempt to answer with this review process is: what are the core and discreet procedural elements of the CESoN and their underpinning evidence?

The following subquestions will structure and focus the data extraction and evidence mapping process:

1. What are the listed discrete elements and core elements outlined within standardised descriptions of the CESoN of the healthy term gestation neonate?
2. What are the recommended techniques and procedural processes for conducting the various elements of the CESoN of the healthy term gestation neonate?
3. What are the differences, similarities and variations in suggested practice between standardised descriptions of the CESoN of the healthy term gestation neonate?
4. What types and quality of evidence underpin recommended best practice of the standardised descriptions of CESoN of the healthy term gestation neonate?
5. What are the omissions and procedural errors in the conduct of the CESoN of the healthy term gestation neonate?

This scoping review is registered with the Open Science Framework: https://osf.io/tujsn.

## METHODS AND ANALYSIS

Leading experts in both scoping review and mapping review methodologies have clarified key differences between these two methodologies[10 11] though it is acknowledged that both types of review have many similarities and overlapping objectives.[12]

For the creation of a systematic evidence map, we are referring to the Campbell Collaboration recommendations,[13] which cautions researchers that evidence maps '…*summarise what evidence there is, not what the evidence says'*.

The review objectives also require us to determine the recommended techniques for the conduct of the various procedural elements of the CESoN. Omissions and errors specific to the conduct of the CESoN will also be identified. This goes beyond the above-stated evidence mapping review objective of summarising the types of evidence. Therefore, this review must also scope the literature to permit an in-depth analysis of the evidence related to practices and procedural elements specific to the CESoN.

This format of this protocol follows both the author guidelines of the target publication for this protocol[14] and is informed by the recommendations[15] on scoping review protocol items to be reported, and the Preferred Reporting Items for Systematic Review and Meta-Analysis Protocols Checklist (PRISMA).[16]

The completed review will be reported using the PRISMA Extension for Scoping Reviews checklist[17] (see online supplemental file 2 appendix 2).

## Eligibility criteria

This review is scoping in nature and seeks to map the evidence underpinning the procedural elements of a multifaceted physical evaluation and screening test performed on all babies within a specific time frame following birth. The procedural elements of the CESoN are not the type of diagnostic tests (observation, inspection, palpation of the baby's body parts and organ systems) that can be evaluated in effectiveness studies. Therefore, the Population, Intervention, Comparator, Outcome, Study Type mnemonic is not suitable to frame the research question and determine eligibility criteria. Following the recommendations of seminal methodology papers[10 11 15 18] including the scoping review chapter of the Joanna Briggs Institute (JBI) Manual for Evidence Synthesis,[19] the PCC mnemonic—Participants (P), Concept, (C) and Context (C) is used to define the inclusion criteria for evidence sources. Eligibility criteria must be congruent with the focus and scope of the review question.

### Participants (P)

Inclusion criteria for neonates being examined will be a gestation of 37+0 weeks or greater at birth; being born with APGAR score greater than seven at 5 min following birth; being primarily cared for by their parent from birth until their CESoN.

Inclusion criteria for practitioners performing the complete examination and screening of the healthy term gestation neonate include paediatricians, neonatologists, midwives, nurses, general practitioners and clinicians or academics or researchers or experts from those professional disciplines that routinely conduct detailed physical examinations of healthy term gestation neonates.

Exclusion criteria for neonates will include neonates born at less than 37 weeks gestation; neonates admitted to a neonatal unit for ongoing care; neonates born with significant congenital disorders or illness diagnosed antenatally or at birth.

Exclusion criteria for practitioners will include pathologists, ultrasonographers and other specialists within fetal medicine, as diagnostic imaging or other investigations used for examination and screening of either the fetus or neonate following stillbirth or neonatal death are not relevant.

## Concept (C)

The concept is the recommended best practices of the procedural elements of the complete examination and screening of the healthy term gestation neonate.

Inclusion criteria for concept: Complete descriptions of the CESoN or the Newborn Infant Physical Examination (NIPE) will be sought. In addition, descriptions of key elements or groups of steps, for example, cardiac screening, developmental dysplasia of the hips (DDH) will be sought, provided they are conducted within the context of the CESoN or the NIPE.

Evidence underpinning the complete descriptions of the CESoN or the NIPE will be sought. In addition, evidence underpinning the description of key elements or groups of steps for example cardiac screening, DDH screening will be sought, provided such evidence is collected within the overall context of the CESoN or the NIPE.

## Context (C)

The context includes the timing and location of the examination. The CESoN may be conducted by a qualified healthcare practitioner in a hospital, primary care or home setting. The CESoN is a distinct and separate physical examination to the less detailed physical examination of the baby completed within an hour of birth which is typically performed by a midwife.[5 20] Evidence will not be drawn exclusively from high-income or English-speaking countries as various procedural elements of the CESoN are conducted by healthcare practitioners from every geographical region.[20]

## Search strategy

An initial scoping search using Google Scholar, Google Searching and OneSearch[21] via the University's library search engine was used to iteratively identify relevant keywords and surrogate terms for the core concepts of the search strings using the PCC framework.

The search strategy for the proposed scoping review and evidence mapping will aim to locate both published and unpublished documents using a three-step search strategy. This will involve an initial limited search of both PubMed and CINAHL to identify studies relevant to the topic. Following this initial search, analysis of the text words contained in the titles and abstracts of relevant articles, and the index terms used to describe the articles further informed the development of a full search strategy for PubMed (see online supplemental file 2 appendix 3).

A second search will then be conducted across all included databases, incorporating all identified keywords and index terms. The databases to be searched include PubMed, CINAHL (EBSCO), Web of Science Core Collection, Scopus, Maternity and Infant Care (MIDIRS), Cochrane Database of Systematic Reviews, Cochrane Central Register of Controlled trials (CENTRAL) and Joanna Briggs Institute Library.

Next, the reference lists of all identified studies and reports will be screened for additional sources. Sources of unpublished and grey literature, Open Grey and Google Scholar will also be searched using an abridged version of the original search strings. Similarly, particular websites of relevant expert academic and professional societies will be searched to locate updated national guidance on specific aspects of newborn screening for example, American Academy of Ophthalmology, American Academy of Orthopaedic Surgeons, American Urological Association, Association for European Paediatric and Congenital Cardiology, Royal College of Paediatrics and Child Health. The key grey literature search terms include newborn and physical examination. In websites or sources where there is a large return of citations, the author will use content relevancy tools, where available and will screen the first 100 citations for potential inclusion.

A time-limit of 10 years will be applied to assessing eligibility for inclusion. Only evidence published in the English language will be included.

## Types of evidence sources

Original research, evidence syntheses, clinical practice guidelines and guidance published by standard-setting organisations or expert professional bodies. In addition, guidance published within comparable healthcare systems in English-speaking countries, the UK, Ireland, Australia, NZ, Canada, the USA, and guidance published in the EU in the English language will be included.

Select core textbooks published or reissued from January 2018 with complete descriptions of the procedural elements of the CESoN will be included.

In addition, literature relating to the effectiveness of steps (published within the context of the CESoN) within the procedure at diagnosing anomalies; or detailing errors or omissions in performance of the procedure will also be included.

## Data charting and synthesis
### Coding and data extraction

A key characteristic of evidence mapping is predefining coding categories ahead of the data extraction process.[22] The predefined coding categories reflect the information required to address the main objectives of the review question. The characteristics of the studies/papers (e.g., source, author, year, country), primary study design and

publication type, participant details, evidence underpinning the development of a description of the CESoN concept studied, context of CESoN and key findings relevant to the review questions will be extracted (see online supplemental file 2 appendix 4).

The data extraction form will be piloted on a sample of three studies by two review authors to assess the feasibility and robustness of the data extraction process. The outcome of the pilot phase will determine whether the data extraction form requires further modifications to comprehensively address the research questions. Any deviations from the above will be outlined in the full scoping review report. Data extraction will be performed by one reviewer (LMG) and then checked by a second review author (ROC or MM). Other review authors will check random samples of data (20% of total papers included) to ensure high-quality data extraction has been achieved. Any disagreements will be resolved by the involvement of a third review author (JH).

The predefined coding categories are the 'framework' to classify the data for the evidence synthesis. As our review question and objectives do not align with the frameworks applied to standard evidence mapping reviews for example, 3ie gap maps, Sightsaver 3 EGMs, we will develop a framework based on the procedural elements of the CESoN. These elements are an amalgamation drawn from three institutions—the American Academy of Pediatrics, the National Institute for Health and Care Excellence and Public Health England. These internationally recognised expert institutions have published recommendations on the CESoN.[1–3] Initially, the procedural elements and associated omissions and errors extracted from the included papers are mapped to the three broad themes:

1. Preparation of the neonate for the examination.
2. Details for the systematic process and procedural steps for the physical examination and screening.
3. Documentation, sharing details of the findings of the examination, providing advice and information.

Comparisons are drawn across included papers in terms of the number of steps, sequencing of steps and errors identified. Once all data are extracted, descriptions of each step are compared and condensed. The aim for each step of the process is to iteratively create a summative, salient and essence-capturing step with a clear beginning and end to a step outlined, based on the comparison of descriptions provided. Errors associated with each step are also summarised.

In order to address the certain research subquestions (1, 4), the data extraction coding framework will also include a mapping of the evidence underpinning recommended best practice for discrete steps or groups of steps relating to discrete assessments conducted as part of the overall CESoN. The evidence underpinning a step or a group of steps is extracted and summarised. Heat maps will be created as cross-tabulations of categorical variables, showing:

1. The number of papers itemising each procedural element (ie, key step or groups of steps).
2. Countries/regions itemising each procedural element (ie, key step or groups of steps).
3. Volume or strength of evidence for each procedural element (key step or group of steps) of the CESoN.

A narrative summary will accompany the tabulated and/or charted results and will describe how the results relate to the reviews objective and question/s.

## Quality assessment

For the traditional scoping review, an assessment of the strength of the body of evidence using the Grading of Recommendations, Assessment, Development and Evaluation tool is not ordinarily conducted.[15]

Seminal papers on scoping review and mapping review methodologies state that critical appraisal of included studies is optional.[10 11] However, other notable papers on mapping review methodology[13 22] recommend critical appraisal of selected included studies to increase confidence in the findings of the evidence synthesis, for example, systematic reviews, primary studies.

For papers which describe the complete CESoN or NIPE, the review authors will describe the processes which underpinned the development of the description or guidance outlining the procedural elements and steps of the CESoN.

The quality of evidence underpinning the elements of the CESoN will be assessed, where possible, using quality assessment tools appropriate to each evidence source that is, primary research, secondary research sources. For example, the quality of any systematic reviews of certain elements of the CESoN (eg, physical examination and screening of the hips for DDH) will be assessed using the JBI critical appraisal tool for systematic reviews.[23] Where a clinical practice guideline is identified, it will be assessed for methodological quality using appraisal tools such as the Critical Appraisal Skills Programme (CASP) checklists and the Appraisal of Guidelines for REsearch & Evaluation (AGREE II) checklist.[24] For expert text and opinion sources, the JBI critical appraisal tool will be used.[25]

## Dissemination

The results from this study will be disseminated through peer-reviewed processes, for example, conference proceedings, peer-reviewed healthcare journals. The completed scoping review will also be included in the doctoral thesis of the lead author (LMG) and made publicly available via CORA - the online open access institutional repository of University College Cork.

**Acknowledgements** I wish to thank Dr Lloyd Philpott for his invaluable support and contributions since joining my doctoral supervision panel in December 2023. A very special acknowledgement and sincere gratitude to my supervisor and co-author Dr Rhona O'Connell upon her recent retirement from University College Cork. This review is to contribute towards a doctoral degree award for author LMG.

**Contributors** LMG, RO'C, MM and JH are involved in conception and design of review, acquisition and analysis of data, interpretation of findings, drafting and reviewing manuscripts.

**Funding** This research received no specific grant from any funding agency in the public, commercial or not-for-profit sectors'. The doctoral studies of the lead author (LMG), of which this work is in part-fulfilment, are funded by their employer (University College Dublin).

**Competing interests** None declared.

**Patient and public involvement** Patients and/or the public were not involved in the design, or conduct, or reporting, or dissemination plans of this research.

**Patient consent for publication** Not applicable.

**Ethics approval** Ethical approval is not required for this scoping review and systematic evidence mapping.

**Provenance and peer review** Not commissioned; internally peer reviewed.

**Data availability statement** The results from this study will be disseminated through peer-reviewed format, that is, conference proceedings, peer-reviewed healthcare journals. The completed scoping review will also be included in the doctoral thesis of the lead author (LMG) and made publicly available via CORA - the online open access institutional repository of University College Cork.

**ORCID iD**
Liz M Greene http://orcid.org/0000-0002-4952-6834

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
