## [Reviewer comments · BMJ Paediatrics Open]

ARTICLE DETAILS

TITLE (PROVISIONAL)	The procedural elements of the complete examination and screening of the healthy term neonate: a protocol for a scoping review and evidence map.
AUTHORS	Greene, Liz M O'Connell, Rhona Murphy, Margaret Hegarty, Josephine

VERSION 1 – REVIEW

REVIEWER	Dr. karel allegaert KU Leuven development and regeneration University Hospitals Leuven Neonatal intensive care unit Herestraat 49 Leuven 3000 Belgium
REVIEW RETURNED	12-Apr-2024

GENERAL COMMENTS	I have read this paper with interest, and a background on clinical neonatal care with an European perspective. My first comment related to the 'link' between the title and the aim of this paper. I'm not a native English person, but do not really understand the 'Procedural elements' of the title, and how did links to 'As a precursor to modifying and adapting an instrument that measures the quality of performance of the CESoN by healthcare professionals, this review aims to identify, evaluate, synthesise, and map the evidence and theory underpinning current practice and procedural elements of the CESoN., as this also covers current practices (topics or aspects to be assessed ?) With a background on clinical care, it is not clear to me to what extent the evolving 'technical' aspects (eg hip screening with ultrasound, structurally implemented in some countries; bilirubin check before discharge – skin or blood; neonatal cardiac screening). Do I understand it correct that you intend to cover both history taking, clinical exams and 'other' techniques as part of this protocol (as you do stress the clinical exam aspect in the CESoN). Somewhat related to this, how will different 'practices' throughout the world be considered, or do you focus on specific territories ? This ambiguity is also somewhat reflected in the supplementary documents (appendix 1). Is the threshold of 37 weeks not somewhat 'high', as the overall majority of 36 weekers are not admitted to a specific neonatal service ? What 'level' of evidence will be considered, as rct will very likely be rather rare, except for eg the more recently introduced technical aspects ?
--

	Does this also covers 'adaptation' syndrome detection.
--	--

VERSION 1 – AUTHOR RESPONSE

Dear Editor(s) and Reviewer - Dr. Karel Allegaert, KU Leuven, Erasmus MC Sophia

Many thanks for providing us with the opportunity to address the comments and feedback items detailed in the Decision Letter dated 17/04/2024.

We hope the revised manuscript and supplementary file have addressed all queries and suggestions for improvement and clarification. Our responses to your comments are detailed in the below table, and also included as 'comments' to the relevant highlighted sections of the manuscript.

Many thanks in advance for your patience as you give so generously of your time to review this updated and revised manuscript.

Kind Regards.
WW, XX, YY, ZZ.

1. Could you please revise the title- which is a bit clunky.
The title has been edited to be more focused.

2. ...clarify the context- will you be mainly drawing upon evidence from high income countries, English speaking?
The context in relation to where evidence will be drawn from and English language has been clarified.

3. ... in the Abstract- clarify that the newborn examination can be carried out by a health professional such as nurse, midwife or physician.
This has been changed as requested to clarify that the newborn examination can be carried out by a health professional such as nurse, midwife or physician.

4. My first comment related to the 'link' between the title and the aim of this paper. I'm not a native English person, but do not really understand the 'Procedural elements' of the title, and how did links to 'As a precursor to modifying and adapting an instrument that measures the quality of performance of the CESoN by healthcare professionals, this review aims to identify, evaluate, synthesise, and map the evidence and theory underpinning current practice and procedural elements of the CESoN., as this also covers current practices (topics or aspects to be assessed ?).

Thank you for this question.

In the first paragraph of the main body, the concept 'procedural elements' has been clarified.

The instrument that we will be modifying and adapting contains items that describe the 'optimal' conduct of the procedural elements i.e., the clinical performance components of the CESoN by the HCP.

However, that instrument was first developed in the early 2000s and it is now out of date. It is outdated because it does not include additional procedural elements that have since been added to the routine CESoN and are now considered standard practice in many jurisdictions e.g., specifying the timing of the CESoN, measurement of head circumference and body length; visual inspection to out rule cleft palate; identifying and managing jaundice; post-ductal oxygen saturation screening.

Therefore, the scoping review will review and map the published evidence so that we can ascertain the recommended techniques for conducting the additional procedural elements that need to be included in the updated version of the instrument.

5. With a background on clinical care, it is not clear to me to what extent the evolving 'technical' aspects (eg hip screening with ultrasound, structurally implemented in some countries; bilirubin check before discharge – skin or blood; neonatal cardiac screening). Do I understand it correct that you intend to cover both history taking, clinical exams and 'other' techniques as part of this protocol (as you do stress the clinical exam aspect in the CESoN).

This scoping review will focus on mapping the descriptions of the procedural (technical) elements of the CESoN. For the healthy term gestation neonate being cared for by their mother, the CESoN procedure typically takes 15 minutes for the HCP (midwife/nurse/physician) to complete.

Our starting point is defining the routine CESoN for healthy term gestation babies based on certain guidelines, specifically the recommendations on examination and screening of the neonate prior to discharge from maternity services to home from many standard setting organisations including Public Health England¹, the

National Institute of Health and Care Excellence², American Academy of Pediatrics³, Canadian Pediatrics Association⁴, and Health Service Executive⁵.

From the above starting point, we can then determine the extent of mapping needed for the evolving technical aspects that you mention. Using your example of hip screening with ultrasound, this is an example of a screening method that is typically conducted by a specialist following the initial hip screening procedural elements in the CESoN (i.e., the examiner assesses for risk factors for DDH, then conducts the clinical hip examination including Barlow and Ortolani manoeuvres). This review will focus on the aspects of history taking and clinical examination techniques specific to the CESoN. We are not mapping the evidence for what could be defined as 'routine postnatal baby care' that is being provided as the baseline newborn care and monitoring outside the CESoN.

6. Somewhat related to this, how will different 'practices' throughout the world be considered, or do you focus on specific territories?

We will be mapping the evidence for the CESoN procedural elements that were published in the English language. We are not focusing on specific territories. We acknowledge that the CESoN as defined by PHE¹, NICE², AAP³, CPS⁴, and HSE⁵ contains procedural elements that may not be standard practice in lower income countries/territories e.g., bilirubin checking with a bilirubinometer. However, the scoping review can map evidence in relation to, for example, techniques for the clinical assessment of black, brown, and yellow skin toned babies (i.e., non-white babies) to assess for jaundice. By not limiting our inclusion criteria to high-income countries/territories, relevant evidence from lower income countries can be mapped.

7. This ambiguity is also somewhat reflected in the supplementary documents (appendix 1).

Thank you for this observation. Appendix 1 sets out the procedural elements of the routine CESoN, as per the guidelines of three internationally recognised standard-setting organisations – Public Health England¹, the National Institute of Health and Care Excellence², American Academy of Pediatrics³.

We have not based the definition of the routine CESoN on just one set of guidelines because there are subtle differences and gaps in each. We consider the use of all three guidelines necessary to define the 'core' recommendations for the routine CESoN, and this permits a broader swathe of high-quality consensus-based evidence from standard setting organisations (including CPS⁴ and HSE⁵) to be considered. It is hoped that a broader definition of the 'core' elements of the CESoN increases the relevance of this scoping review to an international readership of healthcare professionals.

8. Is the threshold of 37 weeks not somewhat 'high', as the overall majority of 36 weekers are not admitted to a specific neonatal service?

Yes, we do acknowledge that many babies born at 36 weeks gestation are not admitted to a neonatal service for specialist care.

However, our judgement was that, given the already large and complex task of mapping the procedural elements of the routine CESoN, we did not wish to create any ambiguity in relation to the neonate. By focusing on the accepted definition of term gestation from 37+0 completed weeks onwards, and also stating a focus on healthy babies being cared for primarily by their mother, we can maintain a focus on mapping the procedural elements for babies that have no pre-existing disorders linked to prematurity.

An additional factor to consider is that we are focusing on the CESoN as a procedure that can also be conducted by specially trained midwives. As a rule, midwives do not conduct the CESoN on babies that fall outside their scope of practice i.e., babies born preterm, or babies with pre-existing congenital disorders.

9. What 'level' of evidence will be considered, as rct will very likely be rather rare, except for eg the more recently introduced technical aspects?

Yes, you are correct that we do not expect to encounter many RCTs in relation to descriptions of the procedural elements of the CESoN. For this reason, the levels of evidence source we will need to consider include expert opinion/consensus from professional and standard-setting bodies, medical textbooks relevant to the CESoN, and clinical practice guidelines. Backward citation mining will be employed to locate the original evidence source for a described clinical procedural technique. If that evidence was published on or after 2013, it will be included in the data charting and synthesis process.

10. Does this also covers 'adaptation' syndrome detection.

We assume that you are referring to 'poor neonatal adaptation' associated with prenatal exposure to Selective Serotonin Reuptake Inhibitors (SSRIs)?

Since this scoping review is focusing on the routine CESoN procedural elements for healthy term gestation neonates, we have not specifically included the term 'adaptation syndrome detection' in the searches of the electronic databases.

The CESoN already includes asking the mother about the perinatal history including medicines taken during pregnancy, as well as examination and assessment of the neurological, gastrointestinal, and respiratory

systems. Therefore, the signs and symptoms of adaptation syndrome are detectable by the routine CESoN. Should a constellation of signs and symptoms suggesting previously unknown adaptation syndrome be detected during the routine CESoN, the midwife/nurse/junior doctor would then refer the baby to a senior or specialist physician for review and follow-up.